# Knowledge Distillation with Multi-granularity Mixture of Priors for Image Super-Resolution

**Simiao Li**[*]
Huawei Noah's Ark Lab

**Yun Zhang**[*]
The Hong Kong University of Science and Technology (Guangzhou)

**Wei Li**[*†] **Hanting Chen**
Huawei Noah's Ark Lab

**Wenjia Wang**
The Hong Kong University of Science and Technology (Guangzhou)

**Bingyi Jing**
Southern University of Science and Technology

**Shaohui Lin**[‡]
East China Normal University

**Jie Hu**[‡]
Huawei Noah's Ark Lab

## Abstract

Knowledge distillation (KD) is a promising yet challenging model compression approach that transmits rich learning representations from robust but resource-demanding teacher models to efficient student models. Previous methods for image super-resolution (SR) are often tailored to specific teacher-student architectures, limiting their potential for improvement and hindering broader applications. This work presents a novel KD framework for SR models, the multi-granularity Mixture of Priors Knowledge Distillation (MiPKD), which can be universally applied to a wide range of architectures at both feature and block levels. The teacher's knowledge is effectively integrated with the student's feature via the Feature Prior Mixer, and the reconstructed feature propagates dynamically in the training phase with the Block Prior Mixer. Extensive experiments illustrate the significance of the proposed MiPKD technique.

## 1 Introduction

Super-resolution (SR) poses a key challenge in computer vision (CV) (Dong et al., 2015; Liang et al., 2021; Chen et al., 2021), reconstructing high-resolution (HR) images from their low-resolution (LR) versions. In the past decade, the convolutional neural network (CNN) (Dong et al., 2014; Kim et al., 2016; Lim et al., 2017) and the Transformer (Wang et al., 2022c; Zamir et al., 2022; Qiao et al., 2024b;a; Tu et al., 2024; Wang et al., 2023; Zhang et al., 2024) have demonstrated exceptional success for SR. However, deploying these models directly on resource-constrained devices is impractical due to their substantial computational overhead (Zhang et al., 2021b). Therefore, there is increasing attention on model compression techniques for super-resolution (SR) models to enhance their practical deployment.

Knowledge distillation, emerging as an effective model compression method, can significantly reduce computation overload, facilitating the student by transmitting prior knowledge from the competent but resource-intensive teacher model to the compact student model (Zhang et al., 2021a; Luo et al., 2021; Hui et al., 2019; Lee et al., 2020; Liu et al., 2023). In contrast to alternative model compression strategies like pruning (Wang et al., 2021a;b), quantization (Li et al., 2020; Hong et al., 2022),

---

[*]Co-first author.
[†]Project leader.
[‡]Corresponding Author. shaohuilin007@gmail.com, hujie23@huawei.com

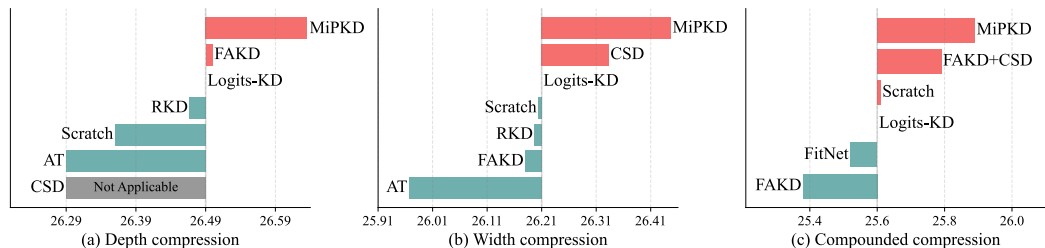

Figure 1: The PSNR values of student models on Urban100 test set under different compression settings. In the depth compression (a), there are barely KD methods outperforming vanilla logits-KD. For width compression (b), CSD performs well but only satisfies this setting. For compounded compression, almost all KD underperforms training without KD.

compact block design (Ahn et al., 2018; Song et al., 2021; Nie et al., 2021; Wang et al., 2022a), and neural architecture search (NAS) (Zoph & Le, 2016; Wan et al., 2020; Ren et al., 2021), KD is a widely recognized method that can be incorporated with these approaches to further enhance the compactness of the student model. KD for SR has also attracted wide attention recently and has gained remarkable progress (Li et al., 2020; Lee et al., 2020; Zhang et al., 2021a; He et al., 2020; Wang et al., 2021b). These methods can be broadly classified into response-based KD and feature-based KD. The former supervises the student model using the teacher model's output, while the latter focuses on aligning the latent space representations between the teacher and student models. (Gou et al., 2021; Wang et al., 2021b; He et al., 2020).

Although previous KD methods show promising results in SR, several issues hinder their wide applications. First, existing KD techniques for SR are tailored to specific teacher-student architectures. They support network depth (Figure 1(a)) or network width (Figure 1(b)) compression (He et al., 2020), and deteriorate the student dramatically when they are adopted into another setting. For instance, FAKD (He et al., 2020) boosts the student model in depth compression but deteriorates the student when applied to a width compression circumstance. CSD (Wang et al., 2021b) improves the student model significantly (Figure 1(b)) but is not compatible with depth compression in Figure 1 (a). It's necessary to propose a more flexible KD framework that is closer to real-world application. While few methods have discussed compounded compression on both depth and width dimensions, which is a much more general but challenging scenario. Existing KD methods for SR, including feature-based methods adapted from high-level CV tasks, such as RKD (Park et al., 2019), AT (Zagoruyko & Komodakis, 2016), and FitNet (Romero et al., 2014), offer limited benefit to the student model. Figure 1 shows that the previous depth and channel distillation methods can just obtain a marginal performance gain or even deteriorate the student in most cases. To alleviate these issues, we introduce a novel knowledge distillation technique for SR models, the multi-granularity Mixture of Priors knowledge for Knowledge Distillation(MiPKD), that is universally applicable to various teacher-student frameworks at feature and block levels. Specifically, the feature prior mixer dynamically combines prior knowledge from the teacher and student models' intermediate feature maps. Then its output-enhanced feature map is supervised by the teacher model's feature map. The block prior mixer adopts a coarser-grained prior mixture at the network block level that dynamically and stochastically switches the normal forward propagation path to the teacher or the student. The output SR image of this ensembled sub-network is supervised by the teacher's output. The primary contributions of this paper are outlined as follows:

- We present MiPKD, a KD framework for efficient SR, transferring the teacher model's dark knowledge from both network width and depth levels. It's flexible and applicable to various teacher-student frameworks.

- We propose the feature and block prior mixers to mitigate the impact of model capacity disparity on the effectiveness of KD to achieve better alignment. The former integrates the feature maps in a unified latent space, while the latter assembles a dynamic combination of network blocks from teacher and student models.

- Extensive experiments on various benchmarks show that the proposed MiPKD framework significantly outperforms the previous arts.

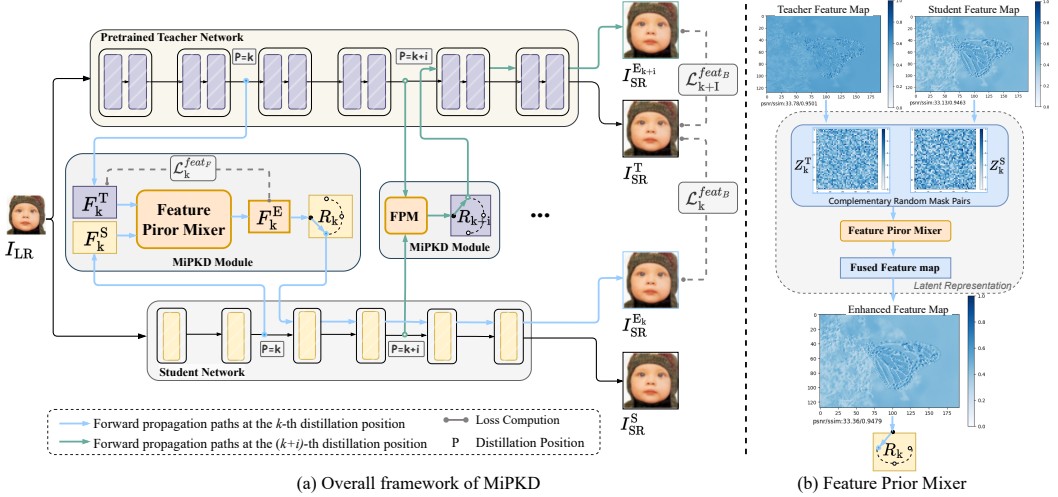

Figure 2: Framework of the MiPKD method. MiPKD utilizes the multi-granularity dark knowledge mixture to constrain the KD process. At the $k$ and $(k+i)$-th distillation position, the Feature Prior Mixer stochastically integrates teacher-provided features with the student model, and the Block Prior Mixer employs a coarser-grained prior fusion at the network block level.

## 2 RELATED WORK

**Efficient SISR.** To improve the model efficiency, there have been various approaches to make the SR model less redundant, such as neural architecture search (NAS) (Chu et al., 2021; Song et al., 2020), pruning (Wang et al., 2021a;b), low-bit quantization (Ma et al., 2019; Li et al., 2020; Hong et al., 2022), and compact net block design (Ahn et al., 2018; Song et al., 2021; Nie et al., 2021; Wang et al., 2022a;c; Zamir et al., 2022). The strength of NAS manifests in searching the optimal architecture but is time-consuming and computationally expensive due to the massive search space. Afterwards, compact SR model designs have attracted rising attention and achieved remarkable progress (Zhang et al., 2022; Hui et al., 2019; Ahn et al., 2018; Dong et al., 2016). ELAN, proposed by Zhang et al. (2022), incorporates the GMSA module that effectively exploits long-range image dependencies and achieves superior performance compared to transformer-based super-resolution models while being much less complex. Pruning (Wang et al., 2021a;b) and quantization (Ma et al., 2019; Li et al., 2020; Hong et al., 2022) are other two types of methods to remove model redundancy by sparsity and low-bit quantization mappings. Despite the considerable progress made by these lightweight networks, significant computational resources are still in demand.

**Feature-based Knowledge Distillation:**

Knowledge distillation is widely recognized as an effective neural network compression technique that is able to significantly reduce the computation overload and improve student's capability by transferring "dark knowledge" as prior information from the large teacher model to the lightweight student model (Gou et al., 2021; Yim et al., 2017; Hinton et al., 2015). Feature-based KD methods extend beyond simple output alignment by focusing on matching intermediate representations. FitNet Romero et al. (2015) aligns feature maps directly, while FAKD He et al. (2020) distills correlation information from the affinity matrix. KD-SRRL Yang et al. (2021) uses a Softmax regression loss by feeding student features into the teacher's classification head. Qiu et al. (2022) introduces a dynamic knowledge mechanism that injects teacher features into the student with hyper-parameters. While MiPKD integrates the teacher's dark knowledge and the student's representations at two levels: feature level and block level. This is achieved through latent space encoding and two stochastic mixing mechanisms, ensuring comprehensive alignment of the learned representations. PEFD Chen et al. (2022) employs a projector ensemble to extract task-relevant discriminative features and avoid overfitting the teacher's feature space. DMAE Bai et al. (2022) aligns intermediate features using a feature-based KD loss and reconstructs the original image via the student's decoder, similar

to FitNet but focused on the direct alignment of intermediate features. ViTKD Yang et al. (2024) combines the FitNet loss for shallow layers with a generative loss for deeper ones.

**Knowledge Distillation for SISR.** Recently, several attempts have also been made for image super-resolution knowledge distillation. Lee et al. (2020) employ a trainable encoder-decoder network to perform information extraction, and use the statistics computed from the scale maps of the decoder to distill student models. He et al. (2020) proposed FAKD to distill the correlation information from the affinity matrix of feature maps. Wang et al. (2021b) proposed CSD that incorporates self-distillation and contrastive learning by introducing extra simply upsampled LR images as negative samples. MTKDSR (Yao et al., 2022) employed two teacher models with different SR objectives (PSNR, perceptual) to guide the student model simultaneously. CrossKD (Fang et al., 2023) divides the teacher and student networks into two segments that are interchanged and connected to perform forward propagation. RDEN (Ren et al., 2024) introduces an efficient SR network design by leveraging re-parameterization and a progressive training strategy. Especially, the FitNet loss is used for distillation in the second training stage. Existing SRKD techniques for SR are tailored to specific teacher-student architectures, focusing on either network depth (Wang et al., 2021b) or channel compression (He et al., 2020), which is infeasible for practical compounded compression applications.

## 3 METHODOLOGY

### 3.1 PRELIMINARIES AND NOTATIONS

Given a low-resolution input image $I_{LR}$, the deep SR model $\mathcal{F}(\cdot)$ aims to reconstruct the high-resolution image $I_{SR} = \mathcal{F}(I_{LR}; \Theta)$ with fine details and consistent content with corresponding high-resolution image $I_{HR}$, where $\Theta$ denotes the model parameters. The logits-based KD method compels the student model $\mathcal{F}_S$ to produce the same output as the teacher model

$$\mathcal{L}_{logits} = \|I_{SR}^T - I_{SR}^S\|_1 \tag{1}$$

where $I_{SR}^S = \mathcal{F}_S(I_{LR}; \Theta^S)$ and $I_{SR}^T = \mathcal{F}_T(I_{LR}; \Theta^T)$ denote the SR images produced by the student and teacher networks. The feature-based KD methods seek to mimic the rich implicit hidden representations between the teacher and the student, which also can be represented by the feature distillation loss

$$\mathcal{L}_{feat} = \|\mathcal{T}_s(F_k^S) - \mathcal{T}_t(F_k^T)\|_1 \tag{2}$$

where $F_k^S$ and $F_k^T$ represent the feature maps of the student and teacher model at the $k$-th distillation position, respectively. $\mathcal{T}_t$ and $\mathcal{T}_s$ are the transformations applied on raw feature maps.

### 3.2 MIXTURE OF PRIOR KNOWLEDGE DISTILLATION

Inspired by MAE (He et al., 2022) that reconstructs the missing pixels from the masked input patches, we proposed the dark knowledge synthesis approach for KD on SR tasks in both feature and block levels. The prior knowledge mixers are applied to the raw feature maps of the student and teacher models in order to encode them into a unified latent space, in which the models' dark knowledge is mixed. Subsequently, the mixed latent feature map is decoded to its original space, enabling the reconstruction of the enhanced feature map and the performance of distillation. While the purpose of the MAE is to reconstruct the masked pixels, the encoder-decoder in the feature prior mixer reconstructs the portion of the teacher model feature map that is replaced by the student's. This allows the student model's intermediate representations to have a similar distribution to the teacher model's. The block prior mixer optimizes the network's capacity to process and represent information. This is achieved utilizing a dynamic combination of blocks, whereby the resulting fusion information is transferred from the feature prior mixer to the enhanced network. The two granularity of prior mixtures follow the common idea of prior mixing and propagation, which effectively mitigates the adverse effects caused by the capacity disparity between the teacher and student.

**Feature Prior Mixer.** Figure 2 illustrates the hybrid dark knowledge framework at the feature level. At the $k$-th feature distillation position, initially, the feature maps of both the student model $F_k^S$ and teacher model $F_k^T$ are processed through their respective encoders to extract the latent representations $\mathbf{Z}_k^S, \mathbf{Z}_k^T \in \mathbb{R}^{C \times H \times W}$ in a unified latent space, where $C, H, W$ represent the dimension of the feature maps. Subsequently, the encoded student and teacher feature maps are fused in accordance with a

pair of randomly generated complementary masks. And the decoder reverts the fused feature map $\mathbf{Z}_k^E$ to the enhanced feature map representation $F_k^E$ in the same space as raw feature maps as

$$F_k^E = \text{Decoder}(Z_k^E) = \text{Decoder}(\mathbf{Z}_k^S \odot (\mathbf{1} - \mathbf{I}^M) + \mathbf{Z}_k^T \odot (\mathbf{I}^M)), \tag{3}$$

where $\mathbf{I}^M \in \{0, 1\}^{C \times H \times W}$ represents a random three-dimensional mask and $\odot$ denotes the element-wise product between matrices. The student's feature map is combined with the teacher's knowledge with the above mixing mechanism to reduce the discrepancy between them at the feature level. $F_k^E$ is utilized as an input to the subsequent block level prior mixer module. The feature distillation loss $\mathcal{L}_k^{feat_F}$ of Feature Prior Mixer is computed between $F_k^E$ and $F_k^T$ as

$$\mathcal{L}_k^{feat_F} = \|F_k^E - F_k^T\|_1 \tag{4}$$

Additionally, in order to enhance the reconstruction capability of the decoder and ensure the stability of training, at the beginning of training, the auxiliary enhanced feature map $F_k^{'E}$ is obtained by directly passing the teacher's feature map to the teacher's encoder and decoder without applying the above masking and mixing strategy. The auxiliary "auto-encoder" loss $\mathcal{L}_k^{ae}$ is computed as

$$\mathcal{L}_k^{ae} = \|F_k^{'E} - F_k^T\|_1 \tag{5}$$

It requires the encoder and decoder to serve as an auto-encoder structure, ensuring the decoded enhanced feature map is comparable with $F_k^T$. The enhancement of the decoder contributes to the overall effectiveness of the feature prior mixer module.

**Block Prior Mixer.** Existing feature-based distillation methods on SR tasks mostly align the feature maps in the original representation space with Mean Absolute Error or Mean Square Error (MSE). The semantic information among the teacher and student networks are differently distributed. Solely aligning features at the present distillation node with the same magnitudes of distance can lead the student model to learn entirely different information. To tackle this issue, we propose to align the networks' ability to process and represent information by assembling a dynamic combination of blocks and transmitting the fusion information from the Feature Prior Mixer to the enhanced network.

To construct an enhanced network ($\mathcal{F}_E^{block}$) at the distillation position $k$, according to the Block Prior Mixing Option $\text{R}_k$ randomly sampled from $\{0, 1\}$, the output of Feature Prior Mixer $F_k^E$ is forwarding propagated to the student network ($\text{R}_k = 1$) or teacher network ($\text{R}_k = 0$), as the propagation path exemplified in Figure 2. The $\mathcal{H}_{S_{(k)}}$ and $\mathcal{H}_{T_{(k)}}$ represent the block from student and teacher models after the current position respectively. $\mathcal{H}_{O_{(k)}}$ represents the mixed block at the current position based on $\text{R}_k$, which can be computed as

$$\mathcal{H}_{O_{(k)}} = \text{R}_k \mathcal{H}_{S_{(k)}} + (1 - \text{R}_k)\mathcal{H}_{T_{(k)}}. \tag{6}$$

Based on this process, denote the output of such concatenated network as $I_{SR}^{E_k}$,

$$I_{SR}^{E_k} = \mathcal{F}_E^{block}(I_{LR}; \Theta^S) = \mathcal{H}_{O_{(k)}}(F_k^E) \tag{7}$$

The feature knowledge distillation loss based on Block Prior Mixer is derived through the combined network's final output with the teacher model's output:

$$\mathcal{L}_k^{feat_B} = \|I_{SR}^{E_k} - I_{SR}^T\|_1 \tag{8}$$

In addition, $\mathcal{L}_k^{feat_B} = 0$ if the $k$-th feature distillation position is dropped out. It is anticipated that there will be an attainment of interchangeability between the corresponding teacher and student network blocks, facilitating the student in learning and replicating the abilities of the teacher model.

**The Whole Pipeline.** Compared to conventional feature-based KD methods, MiPKD uses enhanced feature maps and networks to impose more constraints on the student model. In general, for each feature distillation position $k$, based on the pair of $F_k^T$ and $F_k^S$ as the input of Feature Prior Mixer, the random masked feature maps are fused in a unified latent space. And the $\mathcal{L}_k^{feat_F}$ is computed to align the enhanced feature map $F_k^E$ with the initial teacher feature map in the same representation space. Subsequently, the randomly sampled $\text{R}_k$ determines the propagation option of $F_k^E$, the networks' blocks are randomly exchanged and the knowledge is transmitted from the teacher to student model, as shown in Figure 2. Besides logits-KD loss $\mathcal{L}_{logits}$, reconstruction loss $\mathcal{L}_{rec}$, the feature losses in block and feature levels are accumulated:

$$\mathcal{L}_{total} = \lambda_{kd}\mathcal{L}_{logits} + \lambda_{rec}\mathcal{L}_{rec} + \sum_{k \leq K}(\lambda_{feat}\mathcal{L}_k^{feat_F} + \lambda_{block}\mathcal{L}_k^{feat_B}). \tag{9}$$

Table 1: The specifications for the SR model under ×4 experimental settings, including #Params, FLOPs and FPS are calculated using an input image with dimensions 256×256×3. And Frames per second is evaluated on an NVIDIA V100 GPU.

| Model | Role | Network | | | FLOPs (G) | #Params (M) | FPS |
|-------|------|---------|---|---|-----------|-------------|-----|
| | | Channel | Block | Group | | | |
| EDSR | Teacher | 256 | 32 | - | 3293.35 | 43.09 | 3.2 |
| | Student 1 | 64 | 32 | - | 207.28 | 2.70 | 33.958 |
| | Student 2 | 64 | 16 | - | 129.97 (25.3×) | 1.52 (28.3×) | 53.3 |
| RCAN | Teacher | 64 | 20 | 10 | 1044.03 | 15.59 | 6.3 |
| | Student | 64 | 6 | 10 | 366.98 | 5.17 | 12.3 |
| SwinIR | Teacher | 180 | 6 | - | 861.27 | 11.90 | 0.459 |
| | Student | 60 | 4 | - | 121.48 | 1.24 | 0.874 |

where $\lambda_{kd}$, $\lambda_{rec}$, $\lambda_{feat}$, $\lambda_{block}$ represent the weights for logits-kd loss, reconstruction loss, feature prior mixer and block prior mixer respectively. The teacher's knowledge is effectively transferred through this multi-level distillation process.

## 4 EXPERIMENTAL RESULTS

### 4.1 EXPERIMENT SETUPS

**Backbones and Evaluation.** EDSR (Lim et al., 2017), RCAN (Zhang et al., 2018), and SwinIR (Liang et al., 2021) as backbone architectures are utilized to assess the significance of MiPKD and contrast it against previous KD techniques on ×2, ×3, and ×4 super-resolving scales. The SR network specifications and some statistics are presented in Table 1, including the number of channels, residual blocks and residual groups (RCAN), number of parameters (#Params), FLOPs(Floating Point Operations per Second), and inference speed (frame per second, FPS).

We compare MiPKD with the baselines: train from scratch, Logits-KD (Hinton et al., 2015), RKD (Park et al., 2019), AT (Zagoruyko & Komodakis, 2016), FitNet (Romero et al., 2014), FAKD (He et al., 2020), CrossKD (Fang et al., 2023), and CSD (Wang et al., 2021b). Since the CSD is a self-distillation method in the channel-splitting manner, it's not applicable to the RCAN experiments of network depth distillation. The results for ×4 EDSR trained with CSD are obtained by testing the provided checkpoint, and the ×2 and ×3 ones are reproduced by us since the checkpoints are unavailable. The peak signal-to-noise ratio (PSNR) and the structural similarity index (SSIM) are computed to evaluate the quality of the SR model's output. We utilize DIV2K (Timofte et al., 2017) dataset for training, and evaluate models on various standard test sets: Set14 (Zeyde et al., 2012), Set5 (Bevilacqua et al., 2012), BSD100 (Martin et al., 2001), and Urban100 (Huang et al., 2015).

**Training Details.** The Adam optimizer (Kingma & Ba, 2014) is employed for training models, utilizing parameters $\beta_1 = 0.9$, $\beta_2 = 0.99$ and $\epsilon = 1e - 8$ with $2.5e5$ iterations. The learning rate is initialized at $1e - 4$ and reduced by a factor of 10 at each $1e5$ iteration. We set the loss weights $\lambda_{kd}$, $\lambda_{feat}$ and $\lambda_{block}$ to 1, 1 and 0.1, respectively. The proposed MiPKD is implemented by the BasicSR (Wang et al., 2022b) and PyTorch (Paszke et al., 2019) framework and train them using 4 NVIDIA V100 GPUs. For both training and evaluation, the LR images were produced by applying bicubic down-sampling to the HR images. Augmentation through random cropping, flips, and rotations are applied during training

### 4.2 RESULTS AND COMPARISON

**Comparison with Baseline Methods.** Quantitative results for training EDSR (Lim et al., 2017), RCAN (Zhang et al., 2018), and SwinIR (Liang et al., 2021) of three SR scales are presented in Table 2, Table 3 and Table 11, from which we can draw the following conclusions:

**(1)** Existing KD methods for SR have limited effects, some may even deteriorate the student model. The KD methods originally designed for high-level CV tasks (RKD, AT, FitNet), though applicable,

Table 2: Quantitative comparison of distilling EDSR (Lim et al., 2017) on the benchmark datasets. In these experiments, the EDSR student model of c64b32 is distilled by the teacher model of c256b32.

| Scale | Method | Set5 PSNR/SSIM | Set14 PSNR/SSIM | BSD100 PSNR/SSIM | Urban100 PSNR/SSIM |
|-------|--------|------------|-------------|--------------|----------------|
| x2 | Teacher | 38.20/0.9606 | 34.02/0.9204 | 32.37/0.9018 | 33.10/0.9363 |
|    | Scratch | 38.00/0.9605 | 33.57/0.9171 | 32.17/0.8996 | 31.96/0.9268 |
|    | KD | 38.04/0.9606 | 33.58/0.9172 | 32.19/0.8998 | 31.98/0.9269 |
|    | RKD | 38.03/0.9606 | 33.57/0.9173 | 32.18/0.8998 | 31.96/0.9270 |
|    | AT | 37.96/0.9603 | 33.48/0.9167 | 32.12/0.8990 | 31.71/0.9241 |
|    | FitNet | 37.59/0.9589 | 33.09/0.9136 | 31.79/0.8953 | 30.46/0.9111 |
|    | FAKD | 37.99/0.9606 | 33.60/0.9173 | 32.19/0.8998 | 32.04/0.9275 |
|    | CSD | 38.06/0.9607 | 33.65/0.9179 | 32.22/0.9004 | 32.26/0.9300 |
|    | **MipKD** | **38.18/0.9611** | **33.82/0.9197** | **32.30/0.9011** | **32.56/0.9323** |
| x3 | Teacher | 34.76/0.929 | 30.66/0.8481 | 29.32/0.8104 | 29.02/0.8685 |
|    | Scratch | 34.39/0.927 | 30.32/0.8417 | 29.08/0.8046 | 27.99/0.8489 |
|    | KD | 34.43/0.9273 | 30.34/0.8422 | 29.10/0.8050 | 28.00/0.8491 |
|    | RKD | 34.43/0.9274 | 30.33/0.8423 | 29.09/0.8051 | 27.96/0.8493 |
|    | AT | 34.29/0.9262 | 30.26/0.8406 | 29.03/0.8035 | 27.76/0.8443 |
|    | FitNet | 33.35/0.9178 | 29.71/0.8323 | 28.62/0.7949 | 26.61/0.8167 |
|    | FAKD | 34.39/0.9272 | 30.34/0.8426 | 29.10/0.8052 | 28.07/0.8511 |
|    | CSD | 34.45/0.9275 | 30.32/0.8430 | 29.11/0.8061 | 28.21/0.8549 |
|    | **MipKD** | **34.60/0.9288** | **30.50/0.8454** | **29.21/0.8079** | **28.52/0.8592** |
| x4 | Teacher | 32.65/0.9005 | 28.95/0.7903 | 27.81/0.744 | 26.87/0.8086 |
|    | Scratch | 32.29/0.8965 | 28.68/0.7840 | 27.64/0.7380 | 26.21/0.7893 |
|    | KD | 32.30/0.8965 | 28.70/0.7842 | 27.64/0.7382 | 26.21/0.7897 |
|    | RKD | 32.30/0.8965 | 28.69/0.7842 | 27.64/0.7383 | 26.20/0.7899 |
|    | AT | 32.22/0.8952 | 28.63/0.7825 | 27.59/0.7365 | 25.97/0.7825 |
|    | FitNet | 31.65/0.8873 | 28.33/0.7768 | 27.38/0.7309 | 25.40/0.7637 |
|    | FAKD | 32.27/0.8960 | 28.65/0.7836 | 27.62/0.7379 | 26.18/0.7895 |
|    | CSD | 32.34/0.8974 | 28.72/0.7856 | 27.68/0.7396 | 26.34/0.7948 |
|    | **MipKD** | **32.45/0.8980** | **28.79/0.7865** | **27.71/0.7400** | **26.46/0.7968** |

hardly improve the SR models over training from scratch. For instance, AT and FitNet underperform the vanilla student models trained without KD among all settings.

**(2)** The presented MiPKD outperforms existing KD methods baselines for model compression. For example, MiPKD outperforms the vanilla student in the most challenging dataset Urban100 in EDSR×2, ×3 and ×4 settings by **0.6 dB**, **0.53 dB**, **0.25 dB** in terms of PSNR, respectively as Table 2 shown. Compared with training from scratch, **0.35 dB**, **0.30 dB**, **0.30 dB** in terms of PSNR are improved, respectively, on Urban100 dataset in RCAN ×2, ×3, and ×4 settings as Table 3 shown.

**(3)** The MiPKD is applicable to the transformer network and able to boost the model's performance. Conventional feature-based KD methods are not directly applicable to the Transformer-type networks, so we compare MiPKD with training from scratch and the response-based KD (Hinton et al., 2015) in the experiments. The results in Table 11 indicate that the MiPKD could improve the transformer SR model by a large margin, further emphasizing its superior performance.

**Visual Comparison**. Figure 3 compares the output of ×4 EDSR models from the Urban100 dataset with various KD methods. For instance, for $img\_047$, MiPKD can reconstruct much better fine details than all baseline works. FAKD are prone to artifacts in the left-bottom of the building and the vanilla student, Logits-KD, FAKD, and FitNet are over-blurred. In contrast, MiPKD alleviates the blurring distortions and reconstructs much more fine-grained structural features. Similar observations are observed across other examples, *e.g.* the characters and anisotropic textures in $img\_073$. The visual analysis aligns with the quantitative findings, highlighting the advantages of MiPKD. Further visual comparisons are supplied in the supplementary section.

**Comparison of training costs:** As shown in Table 4, MiPKD significantly outperforms Logits-KD by 0.12dB PNSR, while with an increase of only 0.38s training time per step. It indicates that our MiPKD achieves the best trade-off between performance and training time.

Table 3: Quantitative comparison on RCAN (Zhang et al., 2018) architecture on the benchmark datasets. In these experiments, the RCAN student model of c64b6 is distilled by the teacher model of c64b20.

| Scale | Method | Set5 PSNR/SSIM | Set14 PSNR/SSIM | BSD100 PSNR/SSIM | Urban100 PSNR/SSIM |
|-------|--------|----------------|-----------------|------------------|--------------------|
| x2 | Teacher | 38.27/0.9614 | 34.13/0.9216 | 32.41/0.9027 | 33.34/0.9384 |
| | Scratch | 38.13/0.9610 | 33.78/0.9194 | 32.26/0.9007 | 32.63/0.9327 |
| | KD | 38.17/0.9611 | 33.83/0.9197 | 32.29/0.9010 | 32.67/0.9329 |
| | RKD | 38.18/0.9612 | 33.78/0.9191 | 32.29/0.9011 | 32.70/0.9330 |
| | AT | 38.13/0.9610 | 33.70/0.9187 | 32.25/0.9005 | 32.48/0.9313 |
| | FitNet | 37.97/0.9602 | 33.57/0.9174 | 32.19/0.8999 | 32.06/0.9279 |
| | FAKD | 38.17/0.9612 | 33.83/0.9199 | 32.29/0.9011 | 32.65/0.9330 |
| | CrossKD | 38.18/0.9612 | 33.82/0.9195 | 32.29/0.9012 | 32.69/0.9331 |
| | **MiPKD** | **38.26/0.9614** | **34.02/0.9210** | **32.35/0.9017** | **32.98/0.9357** |
| x3 | Teacher | 34.74/0.9299 | 30.65/0.8482 | 29.32/0.8111 | 29.09/0.8702 |
| | Scratch | 34.61/0.9288 | 30.45/0.8444 | 29.18/0.8074 | 28.59/0.8610 |
| | KD | 34.61/0.9291 | 30.47/0.8447 | 29.21/0.8080 | 28.62/0.8612 |
| | RKD | 34.67/0.9292 | 30.48/0.8451 | 29.21/0.8080 | 28.60/0.8610 |
| | AT | 34.55/0.9287 | 30.43/0.8438 | 29.17/0.8070 | 28.43/0.8577 |
| | FitNet | 34.21/0.9248 | 30.20/0.8399 | 29.05/0.8044 | 27.89/0.8472 |
| | FAKD | 34.63/0.9290 | 30.51/0.8453 | 29.21/0.8079 | 28.62/0.8612 |
| | CrossKD | 34.66/0.9291 | 30.50/0.8448 | 29.22/0.8082 | 28.64/0.8617 |
| | **MiPKD** | **34.76/0.9299** | **30.61/0.8467** | **29.28/0.8090** | **28.89/0.8658** |
| x4 | Teacher | 32.63/0.9002 | 28.87/0.7889 | 27.77/0.7436 | 26.82/0.8087 |
| | Scratch | 32.38/0.8971 | 28.69/0.7842 | 27.63/0.7379 | 26.36/0.7947 |
| | KD | 32.45/0.8980 | 28.76/0.7860 | 27.67/0.7400 | 26.49/0.7982 |
| | RKD | 32.39/0.8974 | 28.74/0.7856 | 27.67/0.7399 | 26.47/0.7981 |
| | AT | 32.31/0.8967 | 28.69/0.7839 | 27.64/0.7385 | 26.29/0.7927 |
| | FitNet | 31.99/0.8899 | 28.50/0.7789 | 27.55/0.7353 | 25.90/0.7791 |
| | FAKD | 32.46/0.8980 | 28.77/0.7860 | 27.68/0.7400 | 26.50/0.7980 |
| | CrossKD | 32.45/0.8984 | 28.81/0.7866 | 27.69/0.7406 | 26.53/0.7992 |
| | **MiPKD** | **32.58/0.8998** | **28.84/0.7875** | **27.75/0.7418** | **26.66/0.8029** |

Table 4: Training expenses of KD methods for distilling ×2 EDSR model.

| KD methods | Logits-KD | FitNet | FAKD | CSD | MiPKD |
|------------|-----------|--------|------|-----|-------|
| Time (s/step) | 0.49 | 0.56 | 0.56 | 1.18 | 0.87 |
| Urban100 PSNR | 31.98 | 30.46 | 32.04 | 32.26 | 32.56 |

## 5 ABLATION STUDY

To illustrate the significance of the proposed MiPKD method, we conduct detailed ablation studies on RCAN and EDSR networks.

**Ablation on the feature and block prior mixers for MipKD.** There are two fine-grained prior mixer modules in MiPKD, namely, the feature and block prior mixers. Their individual effects are ablated in Table 5. The result shows that employing the feature prior mixers leads to significant performance improvement and the block prior mixer based on it could further boost the student model.

**Ablation on the MiPKD feature prior mixer module.** In the feature prior mixer module of MiPKD, the teacher and student models' feature maps are mapped to the latent space through corresponding encoders, then randomly mixed and stitched. We present an analysis on the encoders in Table 6, comparing MiPKD with 1) removing the encoders, aligning and utilizing the teacher's feature map directly and 2) sharing the encoder between the teacher and student model. Removing the encoders would substantially deteriorate the student model's performance. Due to the different distribution of teacher and student models' feature maps, a shared encoder cannot effectively map them to the same latent space, leading to noisy mixtures. Assigning separate encoders to the teacher and student models yields the best results, indicating that mixing feature priors in the same latent space is necessary.

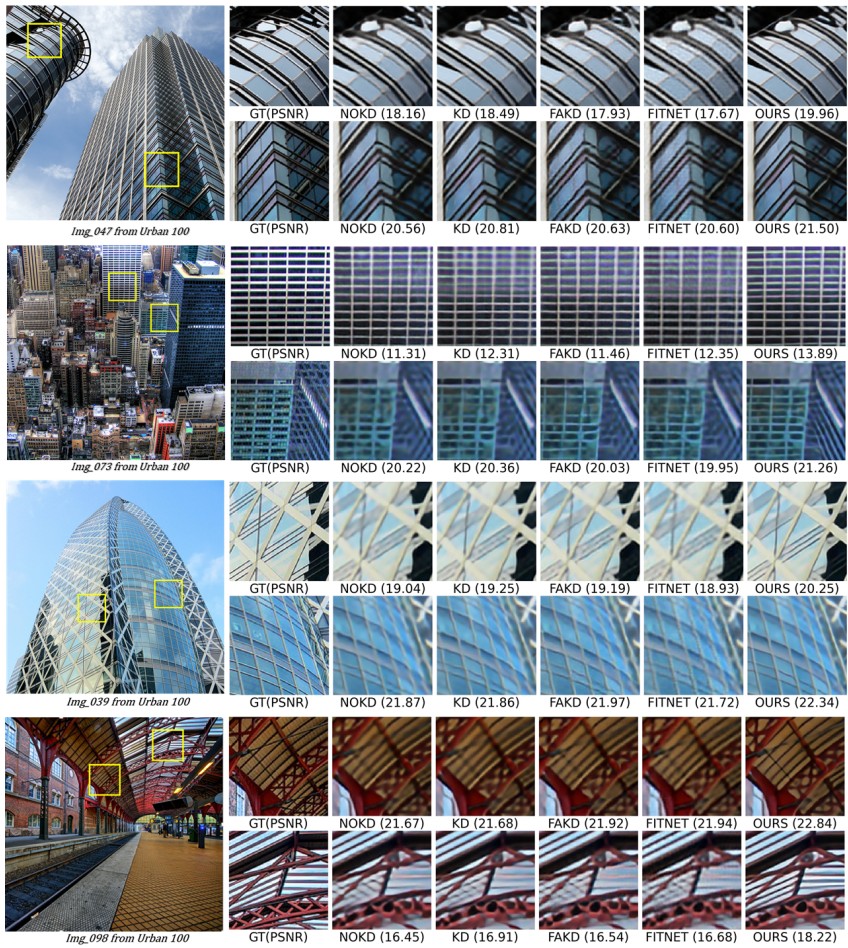

Figure 3: Visual comparison (×4) with existing SRKD methods from Urban100. The numbers in the bracket denote the PSNR of the presented patches.

Table 5: Ablation on the two prior mixers. The RCAN student model of c32b5g5 is distilled by the teacher model of c32b6g10.

| Feature Prior Mixer | Block Prior Mixer | Urban100 PSNR / SSIM |
|:---:|:---:|:---:|
| ✗ | ✗ | 25.60 / 0.7700 |
| ✓ | ✗ | 25.63 / 0.7711 |
| ✗ | ✓ | 25.65 / 0.7717 |
| ✓ | ✓ | 25.69 / 0.7728 |

Table 6: Ablation on the encoder type in MiPKD feature mixer module without block prior mixer module.

| Encoder Type | Urban100 | |
|:---|:---:|:---:|
| | PSNR | SSIM |
| No Encoder | 24.51 | 0.7149 |
| Shared Encoder | 25.61 | 0.7704 |
| Separate Encoder | 25.63 | 0.7711 |

Table 7 compares the encoder and decoder of different network architectures with similar sizes. The convolutional neural network can better project the representations to the unified latent space, as the result shows that CNN exhibits better performance than the MLP encoder/decoder.

**Ablation on the "auto-encoder" loss** $L_k^{ae}$**.** We compared the MiPKD with and without $L_k^{ae}$ in Table 8. The results indicate that the auxiliary "auto-encoder" loss makes the mapping between the raw feature maps' space and the latent space more accurate, leading to a better student model.

Table 7: Comparison of different encoder and decoder network settings.

| Encoder/Decoder Type | Urban100 |
| | PSNR/SSIM |
| --- | --- |
| MLP | 26.42/0.7964 |
| Conv | 26.66/0.8029 |

Table 8: Ablation study on the auto-encoder loss $\mathcal{L}_k^{ae}$.

| Auto-encoder Loss | Urban100 |
| | PSNR/SSIM |
| --- | --- |
| ✗ | 26.42/0.7971 |
| ✓ | 26.66/0.8029 |

Besides, the mask generation strategies are compared in Table 9. Compared with 1) masking according to the Cosine or CKA similarity between teacher and student models' feature maps or 2) generating the complementary pairs of feature map by fixed grid pattern, the random 3D mask exhibits the best performance and least calculation consumption. A more flexible, generalizable strategy is applied in the prior mixer module.

Table 9: Ablation analysis on the masking strategy for feature prior mixture.

| masking strategy | Urban100 PSNR/SSIM |
| --- | --- |
| Cosine | 25.62/0.7711 |
| Grid mask | 25.61/0.7669 |
| CKA | 25.63/0.7713 |
| Random | 25.69/0.7728 |

**Ablation on the Loss weights setting of feature and block mixers.** The impact of various weights of feature mixers loss and block mixer loss is evaluated as Table 10 shown, where $\lambda_{rec}$, $\lambda_{kd}$, $\lambda_{feat}$, $\lambda_{block}$ represent the weights for reconstruction loss, logits-kd loss, feature prior mixer and block prior mixer respectively. Considering the initial fluctuation caused by mixing the block from networks, $\lambda_{block}$ is applied since 0.1 presented the best student performance as the Table 8 shown. In addition, the reconstruction loss of the auto-encoder in the feature prior mixer is introduced in the initial stage of training. As the reconstruction ability of the decoder improves, it's beneficial for the prior mixer to fuse dark knowledge and restore the enhanced feature map efficiently.

Table 10: Ablation analysis on the weights of different losses

| $\lambda_{rec}$ | $\lambda_{kd}$ | $\lambda_{feat}$ | $\lambda_{block}$ | Set5 | Set14 | BSD100 | Urban100 |
| | | | | PSNR/SSIM | PSNR/SSIM | PSNR/SSIM | PSNR/SSIM |
| --- | --- | --- | --- | --- | --- | --- | --- |
| 1 | 1 | 1 | 1 | 32.46/0.8972 | 28.75/0.7851 | 27.68/0.7399 | 26.53/0.7976 |
| 1 | 1 | 0.1 | 1 | 32.34/0.8970 | 28.73/0.7849 | 27.67/0.7394 | 26.47/0.7960 |
| 1 | 1 | 0.1 | 0.1 | 32.42/0.8980 | 28.75/0.7857 | 27.68/0.7399 | 26.51/0.7988 |
| 1 | 1 | 1 | 0.1 | 32.58/0.8998 | 28.84/0.7875 | 27.75/0.7418 | 26.66/0.8029 |

## 6 CONCLUSION

In this paper, we proposed the dark mixing mechanism for KD on SR in feature and block levels. The teacher's knowledge is effectively integrated with the student's feature via the Feature Prior Mixer, and the reconstructed feature propagates stochastically by the Block Prior Mixer. The masked feature maps are fused in a unified latent space, and the mixed prior narrows the optimization space. The Block Prior Mixer propagates the reconstructed feature and re-ensembles the networks to constrain the student model. The two granularity of the prior mixtures follow the common idea of prior mixing and propagation, which alleviates the impact of capacity variation on the performance of knowledge transfer. In-depth evaluations prove the significance of the proposed MiPKD strategy.

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

# A    SUPPLYMENTARY EXPERIMENT RESULTS

## A.1    EXPERIMENT RESULTS ON SWINIR MODEL

We compare MiPKD with other applicable KD methods on distilling transformer-based SR model, SwinIR. The result shows the superiority and universality of MiPKD.

Table 11: Quantitative comparison of distilling SwinIR (Liang et al., 2021) on the benchmark datasets.

| Scale | Method | Set5 | | Set14 | | BSD100 | | Urban100 | |
|---|---|---|---|---|---|---|---|---|---|
| | | PSNR | SSIM | PSNR | SSIM | PSNR | SSIM | PSNR | SSIM |
| 2 | Teacher | 38.36 | 0.9620 | 34.14 | 0.9227 | 32.45 | 0.9030 | 33.40 | 0.9394 |
| | Scratch | 38.00 | 0.9607 | 33.56 | 0.9178 | 32.19 | 0.9000 | 32.05 | 0.9279 |
| | KD | 38.04 | 0.9608 | 33.61 | 0.9184 | 32.22 | 0.9003 | 32.09 | 0.9282 |
| | **MipKD** | **38.14** | **0.9611** | **33.76** | **0.9194** | **32.29** | **0.9011** | **32.46** | **0.9313** |
| 3 | Teacher | 34.89 | 0.9312 | 30.77 | 0.8503 | 29.37 | 0.8124 | 29.29 | 0.8744 |
| | Scratch | 34.41 | 0.9273 | 30.43 | 0.8437 | 29.12 | 0.8062 | 28.20 | 0.8537 |
| | KD | 34.44 | 0.9275 | 30.45 | 0.8443 | 29.14 | 0.8066 | 28.23 | 0.8545 |
| | **MipKD** | **34.53** | **0.9283** | **30.52** | **0.8456** | **29.19** | **0.8079** | **28.47** | **0.8591** |
| 4 | Teacher | 32.72 | 0.9021 | 28.94 | 0.7914 | 27.83 | 0.7459 | 27.07 | 0.8164 |
| | Scratch | 32.31 | 0.8955 | 28.67 | 0.7833 | 27.61 | 0.7379 | 26.15 | 0.7884 |
| | KD | 32.27 | 0.8954 | 28.67 | 0.7833 | 27.62 | 0.7380 | 26.15 | 0.7887 |
| | FitNet | 32.08 | 0.8925 | 28.51 | 0.7800 | 27.53 | 0.7354 | 25.80 | 0.7779 |
| | FAKD | 32.06 | 0.8926 | 28.52 | 0.7800 | 27.53 | 0.7354 | 25.81 | 0.7780 |
| | **MipKD** | **32.39** | **0.8971** | **28.76** | **0.7854** | **27.68** | **0.7403** | **26.37** | **0.7956** |

## A.2    EVALUATE MiPKD ON HIGH COMPRESSION RATE SETTING.

Table 12 shows the results of training ×4 scale EDSR model with both network width and depth compression. The number of parameters is reduced to about $\frac{1}{28}$ of teacher model's. Directly distilling the student model by the teacher model yields negative effects on its performance. Two-stage KD with an intermediate teaching-assistant (TA) model Mirzadeh et al. (2020) are preferred in such case. To make use of the CSD method, we compared different TA options, (1) Teacher->Student1->Student2: the whole pipeline is in a width-then-depth compression order. We adopt CSD to train the TA and FAKD for student model (2) Depth-then-width: we first perform depth compression and then width compression. We adopt FAKD+CSD and MiPKD+MiPKD for the two-stage distillations. The result indicates that distill TA and student model with MiPKD yields the best performance.

Table 12: Quantitative comparison on training ×4 EDSR models with higher compression rate on the benchmark datasets. In these experiments, the EDSR student model of c64b16 is distilled by the teacher model of c256b32.

| Method | Set5 | Set14 | BSD100 | Urban100 |
|---|---|---|---|---|
| | PSNR/SSIM | PSNR/SSIM | PSNR/SSIM | PSNR/SSIM |
| Teacher | 32.65/0.9005 | 28.95/0.7903 | 27.81/0.7440 | 26.87/0.8086 |
| NOKD | 32.01/0.8924 | 28.46/0.7782 | 27.47/0.7324 | 25.61/0.7704 |
| KD | 31.99/0.8921 | 28.46/0.7784 | 27.47/0.7327 | 25.60/0.7700 |
| FitNet | 31.92/0.8912 | 28.42/0.7776 | 27.44/0.7317 | 25.52/0.7672 |
| FAKD | 31.65/0.8879 | 28.32/0.7760 | 27.37/0.7303 | 25.38/0.7629 |
| FAKD+CSD | 32.00/0.8930 | 28.47/0.7800 | 27.51/0.7340 | 25.79/0.7790 |
| CSD+FAKD | 31.86/0.8907 | 28.42/0.7786 | 27.46/0.7327 | 25.58/0.7709 |
| **MiPKD+MiPKD** | **32.17/0.8947** | **28.57/0.7812** | **27.57/0.7354** | **25.89/0.7794** |

## A.3    ABLATION ON THE FREQUENCY OF BLOCK PRIOR MIXER

In the MiPKD, the block prior mixer at each distillation position randomly determines if the enhanced feature map $F_k^E$ is forward propagated to the student or teacher network. In Table 13, we analyzed the impact of frequency of this random propagation. It's also the number of each input image

passing through the enhanced mixed SR networks. It shows that performing single random forward propagation yields the best results, which in practice is the most efficient as well.

Table 13: Ablation for the frequency of random propagation of the Block Prior Mixer. The RCAN student model of c64b5g10 is distilled by the teacher model of c64b6g10.

| #propagation per sample | Set5 | | Set14 | | BSD100 | | Urban100 | |
| :---: | :---: | :---: | :---: | :---: | :---: | :---: | :---: | :---: |
| | PSNR | SSIM | PSNR | SSIM | PSNR | SSIM | PSNR | SSIM |
| 1 | 31.99 | 0.8919 | 28.43 | 0.7777 | 27.46 | 0.732 | 25.69 | 0.7728 |
| 2 | 31.87 | 0.8884 | 28.38 | 0.7632 | 27.42 | 0.7311 | 25.62 | 0.7699 |
| 4 | 31.86 | 0.8867 | 28.37 | 0.7619 | 27.41 | 0.7306 | 25.57 | 0.7697 |

## B    MORE VISUAL RESULTS

In the experiment section of main text, we provided some visual results of EDSR model. We present more visual comparisons of MiPKD with other KD methods over the SwinIR network in Figure 4. MiPKD reconstructs more structural details and alleviates the blurring artifacts, which is consistent with the observations on EDSR network.

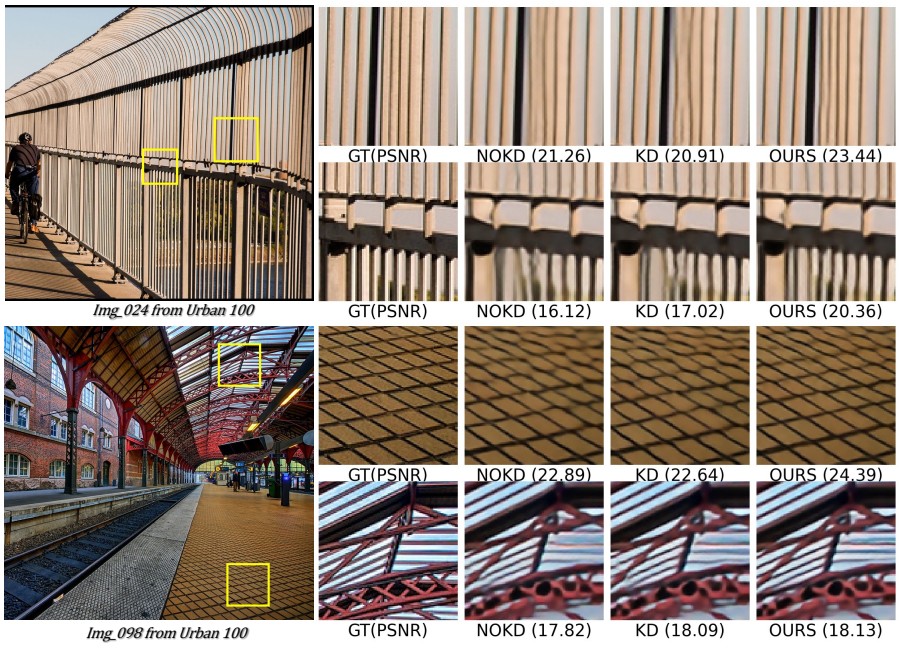

Figure 4: The ×4 super resolution results of SwinIR models on img024, and img098 from Urban100. PSNRs of the cropped regions are annotated below each image.

## C    FEATURE MAP ANALYSIS FOR THE MASK STRATEGY

The comparison of feature maps at a deep layer (the convolution layer following all residual groups of RCAN) across FitNet, RKD, and MiPKD could highlight the effectiveness of MiPKD's random masking strategy, as shown in Figure 5. The MiPKD excels in capturing fine-grained texture details, producing feature maps (far right) with sharper contrasts between high-frequency regions with complex patterns and low-frequency background regions. This enhanced representation serves as higher-quality input for the network's reconstruction layers, ultimately leading to superior visual quality in the output SR images.

The Feature Prior Mixer ensures that the student learns to reconstruct the teacher's fine-grained features while maintaining contextual consistency, effectively narrowing the gap between teacher and student feature distributions. Simultaneously, the Block Prior Mixer introduces stochastic routing of feature representations, enabling the student to inherit complex transformation patterns from the teacher. This two granularity prior mixing mechanism enhances the fidelity of high-frequency textures while preserving structural integrity, reducing artifacts and producing visually pleasing super-resolution results with sharper edges and more realistic textures.

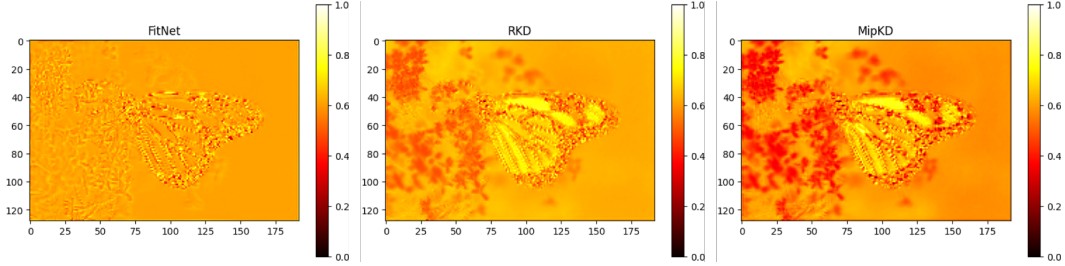

Figure 5: The feature maps at the deep layer ('conv afterbody') distillation point for FitNet, RKD, and MiPKD methods on the ×4 super resolution RCAN model

## D   TRAINING STABILITY OF MIPKD

The MiPKD adopts random masking strategy in the feature and block prior mixer, which introduces diverse mixing patterns. As shown in Figure 6, the random strategy is as stable stable as other feature-based KD methods during training.

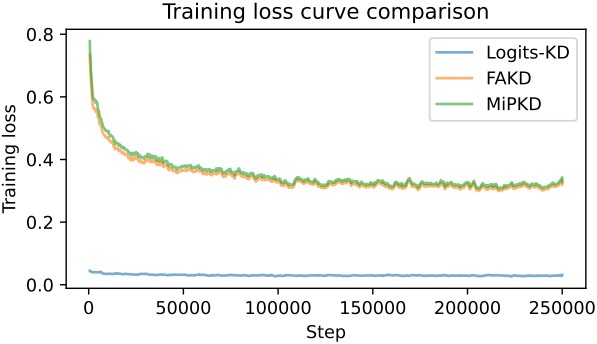

Figure 6: Loss curve of distilling ×4 RCAN with different KD methods.

