# OpenReview forum: "Knowledge Distillation with Multi-granularity Mixture of Priors for Image Super-Resolution"
_ICLR.cc/2025/Conference — ICLR 2025 Spotlight_

### Official Review · Reviewer_qbB2 · 2024-10-27

**Soundness:** 3
**Presentation:** 3
**Contribution:** 2
**Rating:** 8
**Confidence:** 4

**Summary:**

This work proposes a new knowledge distillation method which considers multi-granularity mixturw of priors for image super resolution. Motivated by the success of masked autor encoder in reconstructing missing pixels from masked input, this work explores this machinism into feature prior mixer and block prior mixer to distill teacher's feature. Experiments are done on various datasets to show it effectiveness but the improvement is not significant,

**Strengths:**

1. The proposed method is well motivated and grounded.  This work tackles the challenge of aligning representations between models of different sizes, which is important for improving the performance of lightweight models.
2. Experiments are done on various datasets (Set5, Set14, BSD100, and Urban100) to show its superiority over competitors in super-resolution task.
3. The authors perform ablation studies to evaluate the contribution of each component (feature prior mixer, block prior mixer, encoder type) and discuss different loss weighting strategies.

**Weaknesses:**

1. This method lacks comparison with SOTA feature-based knowledge distillation methods, such as, [1][2][3][4] which all contributed to projecting features before distillation to improve student's performance albeit not specifically in the super-resolution domain.
[1]ViTKD: Feature-based Knowledge Distillation for Vision Transformers
[2]Improved feature distillation via projector ensemble
[3]Knowledge distillation via softmax regression representation learning
[4]Masked Autoencoders Enable Efficient Knowledge Distillers
2. As a closely related work [4] also proposes masked auto encoder for feature distillation.  It would be beneficial to discuss this work in both the introduction and experimental sections, highlighting the differences and demonstrating where the proposed method offers improvements.
Besides, [5] has included knowledge distillation used for super resolution.  Method such as RDEN in [5] should be included in related work.
[5]The Ninth NTIRE2024 Efficient Super-Resolution Challenge Report
3. Three key factors influence Knowledge distillation performance: knowledge, position, loss. Current ablations studies have discusses knowledge and loss but lacks discussion of where to put.  Understanding the optimal positions for applying the feature prior mixer and block prior mixer would provide valuable insights into the flexibility and effectiveness of MiPKD.

**Questions:**

The main concern with the proposed method is the lack of comparison with closely related feature-based knowledge distillation approaches, such as ViTKD [1], Improved Feature Distillation via Projector Ensemble [2], and Knowledge Distillation via Softmax Regression Representation Learning [3]. These methods, while not specifically targeting super-resolution, focus on feature projection before distillation, making them relevant benchmarks. Including these comparisons would provide a clearer understanding of the proposed method’s strengths and positioning relative to state-of-the-art techniques. Additionally, a deeper discussion on the differences between the proposed method and Masked Autoencoders [4] would clarify its unique contributions.   [5] serves as a benchmark for super-resolution where knowledge distillation is applied to enhance performance. Including a discussion of this benchmark and similar methods would help clarify how the proposed method contributes to this area.

**Details Of Ethics Concerns:**

Based on the provided information, there are no ethics concerns regarding this paper.

---

> ### Author Response · Authors · 2024-11-24
>
> We sincerely thank you for your positive evaluation. Your comments are deeply encouraging with valuable insights. If you have any further suggestions or would like to discuss our work in more detail, we would be delighted to engage in a conversation. Your feedback is invaluable to us. Detailed responses are provided in the following rebuttal.
>
> # Weakness 1: Discussion and comparison with more recent SOTA KD methods.
>
> Thank you for highlighting these related works, which are all feature-based distillation methods. Below, we will discuss the difference between our MiPKD and these methods in terms of feature alignment, and provide the experimental comparison on SR task.
>
> 1. **ViTKD** aligns the student’s features with the teacher’s by using the FitNet loss for shallow layers and a generative loss for deeper layers. Its generative KD loss can be seen as a special case of our Feature Prior Mixer in MiPKD, without the auto-encoder structure or stochastic feature mixing.
>
> 2.  **PEFD** employs a projector ensemble to prevent overfitting to the teacher’s feature space, which learns the task-relevant discriminative feature. In contrast, MiPKD employs the feature mixture in the latent space by adding the part feature information as the prior, which ensures better feature learning for student.
>
> 3. **KD-SRRL** computes a Softmax Regression loss by feeding the student’s features to the teacher’s classification head for distillation In contrast, MiPKD leverages a block prior mixer to stochastically passing the enhanced feature maps for the alignment on the final outputs, which effectively reduces the capacity disparity between the teacher and student.
>
> 4. **DMAE** applies a feature-based KD loss to align the intermediate features between the pre-trained MAE teacher model and the student based on the masked image, and employs the decoder after the last student’s layer to reconstruct the original image. Similar to the FitNet approach, it forces direct feature alignment on the intermediate features of teacher and student. In contrast, MiPKD uses an auto-encoder like structure to enable feature prior mixture in the latent space. Moreover, MiPKD employs the random mask on the feature space, instead of the masking on the input images in DMAE. A more detailed comparison between the MiPKD and DMAE is below
>     - The DMAE aligns intermediate features between teacher and student using a simple L1 loss, with lightweight MLPs employed to match their dimensions. In contrast, MiPKD adopts a Feature Prior Mixer to encode teacher and student feature maps into a unified latent space for mixing and reconstruction, further stabilized by an auxiliary auto-encoder loss $\mathcal{L}^{ae}\_k$.
>     - While DMAE does not explicitly distill at the block level, MiPKD introduces a Block Prior Mixer, which dynamically switches between teacher and student blocks for coarse-grained distillation.
>     - In terms of knowledge transfer, DMAE relies on masked image reconstruction to facilitate feature learning, whereas MiPKD mixes feature maps and blocks at multiple granularities, aligning models’ logits, features, and reconstruction outputs to address capacity disparity.
>     - The DMAE is tightly integrated into the MAE framework, requiring the teacher to be pre-trained with masked image modeling. In contrast, MiPKD is universally compatible with various SR architectures and can be extended to diverse computer vision tasks.
>
> Furthermore, we apply these methods to the SR tasks for the experimental comparison. As shown in the below table, our MipKD significantly outperforms all above feature-based distillation methods. For example, we outperform DMAE by 0.19db PSNR for x4 SR, when distilling RCAN model.
>
> | Method            | Urban100 |
> | ----------------- | ----------------------------- |
> | ViTKD [1]         | 26.440/0.799                  |
> | PEFD [2]          | 26.420/0.797                  |
> | KD-SRRL [3]       | 26.450/0.796                  |
> | DMAE [4] (FitNet) | 26.470/0.798                  |
> | MipKD             | 26.660/0.803                  |

---

> > ### Comment · Reviewer_qbB2 · 2024-11-24
> > **Comments from qbB2**
> >
> > Thank you for your reply and for conducting the additional experiments. I believe my concerns have been clarified.

---

> > > ### Author Response · Authors · 2024-11-26
> > >
> > > Dear Reviewer qbB2,
> > >
> > > Thank you for your valuable feedback and constructive comments. We greatly appreciate the time and attention you have dedicated to this process.
> > >
> > > Sincerely Yours,
> > >
> > > The Authors

---

> ### Author Response · Authors · 2024-11-24
>
> # Weakness 2: Discussion with DMAE and RDEN.
>
> The difference between our method and DMAE is discussed in the answer of Weakness 1. For RDEN, it introduces an efficient SR network design by leveraging re-parameterization and a progressive training strategy. Especially, the FitNet loss is used for distillation in the second training stage. As a general distillation framework, MiPKD can serve as a replacement for RDEN’s original distillation method, to potentially further enhance  student performance. In the revision, we have cited this paper and discussed it in the related work.
>
> # Weakness 3: Exploration on the optimal distillation position.
>
> Thank you for raising this important point regarding the positioning of distillation components. We agree that understanding the optimal placement of the feature and block prior mixers can provide valuable insights into the flexibility and effectiveness of MiPKD. As suggested, we conduct additional ablation experiments to evaluate the impact of applying the FPM and BPM at different network positions, which is shown in the below table.
>
> | Distillation position       | RCAN x4 Urban100 |
> | --------------------------- | ---------------- |
> | Early layer (conv_first)    | 26.57/0.8017     |
> | Intermediate layer          | 26.54/0.7996     |
> | Deep layer(conv_after_body) | 26.55/0.7998     |
> | All three above positions   | 26.66/0.8029     |
>
> The results indicate that applying MiPKD uniformly across all layers yields the best performance compared to the specific single layer. To explain, the uniform strategy captures hierarchical feature interactions between the teacher and student models.

---

### Official Review · Reviewer_B9sn · 2024-11-02

**Soundness:** 3
**Presentation:** 3
**Contribution:** 3
**Rating:** 8
**Confidence:** 5

**Summary:**

The authors propose MiPKD, a multi-granularity mixture of prior knowledge distillation framework designed for image super-resolution tasks. MiPKD facilitates the transfer of “dark knowledge” from teacher models to student models across diverse network architectures. The framework employs feature and block prior mixers to reduce the capacity disparity between teacher and student models for effective knowledge alignment and transfer. Extensive experiments are conducted on three SR models and four datasets, demonstrating that MiPKD significantly surpasses existing KD methods.

**Strengths:**

1. The integration of feature fusion within a unified latent space  and stochastic network block fusion are innovative for SR model knowledge distillation.
2. The paper provides clear explanations and discussion, supported by well-organized charts and ablation studies, which effectively highlight the framework’s contributions and innovations.
3. The paper is well organized and the motivation of MiPKD is clear

**Weaknesses:**

1. The writing is sometimes unclear, with inconsistent notations and undefined terms:
	- The feature maps $F$ in Fig. 2, Eq. 2, and Eq. 3 are not consistently bolded.
	- The three feature maps in Eq. 3 are inconsistently formatted.
	- The loss weights $\lambda_1$ and $\lambda_2$, mentioned in lines 358–359, are not defined or referenced elsewhere in the paper.
2. While the paper provides comparisons for EDSR and RCAN, the evaluation on SwinIR is not as comprehensive.

**Questions:**

1. What is the rationale for randomly sampling the forward propagation path in the block prior mixer? If the output of Feature Prior Mixer is passed to both teacher and student models and compute two losses, will the knowledge distillation be more effective?
2. What are the specifications of the encoder and decoder networks (e.g., number of layers, hidden dimensions)? Would a larger auto-encoder result in a better student model?

---

> ### Author Response · Authors · 2024-11-24
>
> We sincerely thank you for your positive evaluation. Your comments are deeply encouraging with valuable insights. If you have any further suggestions or would like to discuss our work in more detail, we would be delighted to engage in a conversation. Your feedback is invaluable to us. Detailed responses are provided in the following rebuttal.
>
> # Weakness 1: Inconsistent notations and undefined terms
>
> We sincerely thank the reviewer for pointing out these issues, which are essential for improving the clarity and consistency of our manuscript. In future version , we will carefully revise the notations, and add explicit description and reference for them in the relevant sections to avoid ambiguity.
>
> # Weakness 2: Experiments on SwinIR model
>
> Thank you for pointing out the limited evaluation on SwinIR. We acknowledge that the coverage for this model is not as extensive as for EDSR and RCAN. The SwinIR model, being based on a Transformer architecture, poses unique challenges for knowledge distillation, as most existing SR KD methods are primarily designed for CNN-based networks. In the experiments, we applied two feature-based KD methods (namely, FitNet and FAKD) on SwinIR, and found that both of them lead to worse student model comparing to training from scratch, while the MiPKD improves SwinIR student model significantly.
>
> # Question 1: Rationale for randomly sampling the forward propagation path.
>
> The block prior mixer dynamically assembles network blocks from the teacher and student models. Randomly sampling the forward propagation path ensures a diverse set of combinations, encouraging the student model to generalize better by learning from various teacher-student configurations. Such a stochastic approach mitigates overfitting and reduces the capacity disparity between teacher and student models. By selectively using teacher or student blocks during propagation, the mixer provides flexibility and enables the student to progressively inherit and adapt the teacher’s capabilities.
>
> # Question 2: Specifications of the encoder and decoder networks.
>
> In the Feature Prior Mixer module, separate encoders are designed for the teacher and student models to ensure effective mapping to the latent space, accommodating the distinct feature distributions of each. The decoders transform the fused feature maps back into enhanced representations for subsequent distillation. The encoder and decoder architectures are deliberately lightweight, focusing on dimensional consistency and efficient projection into the latent space rather than performing deep feature extraction. Typically, three convolution layers are deep enough. Table 8 provides comparisons between MLP and CNN-based encoder/decoder networks.
>
> To address your question on the size of encoder and decoder, we include an ablation study that evaluates the impact of increasing the number of convolutional layers in them.
>
> | Number of Convolution layers in Encoder and Decoder | RCAN X4 Urban100 |
> | ------------------------------------------------------- | ---------------- |
> | 1 | 26.18/0.7885 |
> | 2 | 26.58/0.8006 |
> | 3 | 26.66/0.8029 |

---

> > ### Comment · Reviewer_B9sn · 2024-11-26
> > **Rebuttal**
> >
> > Thank you for your detailed rebuttal. All my concerns are well addressed in this rebuttal. I have also carefully read other reviewers' comments and their responses. The authors have added detailed discussion to analyze the rationale of Feature Prior Mixer (FPM) and Block Prior Mixer (BPM), and provided comprehensive experiments.
> >
> > After careful consideration, I believe it is  a high-quality paper with interesting and novel FPM and BPM. Thus, I will increase my score.

---

> > > ### Author Response · Authors · 2024-11-26
> > >
> > > Dear Reviewer B9sn,
> > >
> > > Thank you for your valuable feedback. We greatly appreciate the time and attention you have dedicated to our paper.
> > >
> > > Sincerely Yours,
> > >
> > > The Authors

---

### Official Review · Reviewer_pGYs · 2024-11-03

**Soundness:** 3
**Presentation:** 2
**Contribution:** 2
**Rating:** 6
**Confidence:** 4

**Summary:**

This paper introduces a multi-granularity prior knowledge distillation (MiPKD) framework for image super-resolution (SR) tasks. By incorporating both a feature prior mixer and a block prior mixer, MiPKD effectively transfers knowledge from a larger teacher model to a compact student model, enhancing model compression while preserving SR performance. Unlike conventional KD methods tailored to specific teacher-student architectures, MiPKD flexibly accommodates different network depths and widths.

**Strengths:**

1. Originality: MiPKD’s innovation lies in its multi-granularity knowledge mixing mechanism. Through coordinated use of feature and block prior mixers, MiPKD enables adaptable knowledge transfer across different teacher-student architectures, a notable improvement in KD design.

2. Quality: Experimental design is thorough and covers a wide array of SR network architectures and compression configurations. Results indicate MiPKD’s superior performance on multiple datasets, confirming its generalizability and efficacy across tasks.

3. Clarity: The paper’s organization is clear, with helpful visuals and an accessible writing style that conveys the method’s complexity effectively.

**Weaknesses:**

1.Explanability. Although MiPKD seems good in experiments, I still don't know why it is effective. The method introduces random masks I and R_k in the feature prior mixture and block prior mixture. but it needs to clarify why these masks are effective in distilling SR networks. I will be appreciated if the author could convince me through feature map analysis or theoretical deductions.

2. Experiment Details: While MiPKD demonstrates good results, further details regarding the impact of different mask generation strategies in the feature mixer could clarify the specific role and contribution of these strategies.

3. Generality: The paper primarily discusses MiPKD’s performance in SR tasks, but it seems that MiPKD is not specially designed for the SR task. Thus, the authors are encouraged to show the applicability to other CV tasks. Additional experiments on a broader range of tasks could enhance the method’s perceived generalizability.

4. Experiments on current SOTA SR networks. Experiments are carried out on EDSR, RCAN and SwinIR networks. If possible, the authors could do experiments on more recent SOTA networks, such as DRCT-L and HMANet etc.

**Questions:**

1. Does the block-level mixing lead to instability in training for certain models? Were there hyper-parameter adaptations needed for different architectures during training?

2. Have the authors considered extending MiPKD to other vision tasks, such as object detection or semantic segmentation, to verify the broader applicability of the proposed framework?

---

> ### Author Response · Authors · 2024-11-24
>
> We sincerely thank you for your positive evaluation. Your comments are deeply encouraging with valuable insights. If you have any further suggestions or would like to discuss our work in more detail, we would be delighted to engage in a conversation. Your feedback is invaluable to us. Detailed responses are provided in the following rebuttal.
>
> # Weakness 1: Why random masks are effective in distilling SR networks?
>
> The masking strategy in MiPKD plays a crucial role by using a stochastic mixing of feature maps from the teacher and student models within a unified latent space, and a stochastic routing between teacher and student blocks to assemble a dynamic combination of blocks through the SR output alignment. In this way, MiPKD effectively reduces the capacity disparity between the teacher and student.
>
> Specifically, a **random 3D masks ($I$)** is used in feature prior mixer (FPM). The encoder-decoder structure in the FPM reconstructs the masked teacher features, facilitating the student's approximation of the teacher's feature distribution. It encourages the student model to align its intermediate representations with the teacher’s by adding the part feature information as the prior, which ensures better feature learning for student.
>
> **Block Prior Mixer with Mask** $R\_k$​: Mask $R\_k$​ introduces stochastic routing between teacher and student blocks, allowing the student to learn from the teacher’s pathways without replicating its structure. This block-level fusion helps the student model adapt to complex spatial and texture variations, gradually learning the teacher’s feature transformation style to enhance the feature presentation ability of the student.
>
> To further support this explanation, feature map analyses are added to the supplementary material as Fig.3 shown, demonstrating how the masking strategy enhances the student model’s representation learning and overall performance. The random masking approach in MiPKD encourages the student model to focus selectively on critical and diverse features from the teacher's output rather than attempting a full replication. This combination of random masking allows MiPKD to effectively balance global context with detailed textures, making it particularly well-suited for high-detail tasks, such as super-resolution.

---

> ### Author Response · Authors · 2024-11-24
>
> # Weakness 2: More discussions on the mask generation strategies.
>
> MiPKD employs a random 3D mask as the default masking strategy in the Feature Prior Mixer. This approach selectively combines feature maps from the teacher and student models in the latent space, introducing the controlled stochasticity and diversity into the mixing process. It encourages the student model to learn essential teacher features across varied spatial configurations.
>
> In Tab. 10, we compared the random 3d mask generation strategy with two alternatives: **grid masking** and **similarity-based masking** (by Cosine and CKA similarities). Grid masking applies the predefined grid pattern to select the masking position, while similarity-based masking guides the mixing based on feature similarity. The ablation study indicates that the random 3D mask achieves the best performance, compared to other strategies.

---

> ### Author Response · Authors · 2024-11-24
>
> # Weakness 3, Question 2: Extension to other vision tasks
>
> Both Feature Prior Mixer and Block Prior Mixer strategies are flexible for different network architectures, which has strong potential for extension to other vision tasks. As such, we add the experiment by applying MiPKD to the ImageNet-1k classification task as an auxiliary and standard task, as shown in Table below. By distilling the ResNet18 model using  the ResNet34, MiPKD still outperforms feature distillation method SRRL[1] by 0.15 Top-1 accuracy. The multi-granularity prior mixing helps the student model capture crucial high-level semantic information, which potentially verify the broader applicability of the proposed MiPKD.
>
> | Method            | Teacher | Student | KD    | AT    | SRRL  | MiPKD |
> | ----------------- | ------- | ------- | ----- | ----- | ----- | ----- |
> | top-1 accuracy(%) | 73.31   | 70.04   | 70.68 | 70.59 | 71.73 | 71.88 |
>
>
> [1] Yang, J., Martínez, B., Bulat, A., & Tzimiropoulos, G. (2021). Knowledge distillation via softmax regression representation learning. ICLR

---

> ### Author Response · Authors · 2024-11-24
>
> # Weakness 4: Experiments on more recent SOTA networks
>
> As suggested, we apply the proposed MiPKD to distill the DRCT[2] model for x4 SR, as shown in the below table. The proposed MiPKD outperforms the existing KD methods, including Logits-KD and FitNet. For example, MiPKD outperforms the logits KD by 0.13 dB for x4 SR on Urban100.
>
> | Model              | Teacher(27.6M)| Student(14.1M)| Logits-KD    | FitNet       | MipKD        |
> | ------------------ | ------------- | ------------- | ------------ | ------------ | ------------ |
> | Urban100 PSNR/SSIM | 28.78/0.8492  | 27.23/0.8188  | 27.28/0.8195 | 27.27/0.8190 | 27.41/0.8221 |
>
> [2] Hsu, C., Lee, C., & Chou, Y. (2024). DRCT: Saving Image Super-Resolution away from Information Bottleneck. 2024 CVPR

---

> ### Author Response · Authors · 2024-11-24
>
> # Question 1: Concern on the training stability and hyper-parameter optimization
>
> In our experiments, we followed the standard hyper-parameter setting (learning rate, lr scheduler, and optimizer), as summarized in section 4.1. For the loss weights $\lambda\_{rec}, \lambda\_{kd}, \lambda\_{feat}, \lambda\_{block}$, we have evaluated their effect in Tab. 11. We further add the ablation about the distillation position of MiPKD on RCAN as shown in the table below. Applying MiPKD uniformly at early, intermediate and deep layers achieves the best performance. For simplicity, we apply the setting of all positions for other models, which also achieves the high distillation performance. The training process of MiPKD is stable to distill SR models. For example, the training curve of X4 RCAN network is added in Fig.4 of the supplementary material to demonstrate the training stability.
>
> | Distillation position                                       | RCAN x4 Urban100 |
> | ----------------------------------------------------------- | ---------------- |
> | Early layer: first convolution layer                   | 26.57/0.8017     |
> | Intermediate layey: the 3rd RCAN residual groups        | 26.54/0.7996     |
> | Deep layer: convolution layer after all residual groups | 26.55/0.7998     |
> | All three positions                                         | 26.66/0.8029     |

---

> > ### Comment · Reviewer_pGYs · 2024-11-25
> >
> > Thank you for your explanation and extra experiments, which are very helpful for me to understand the article

---

> ### Author Response · Authors · 2024-11-26
>
> Dear Reviewer pGYs,
>
> Thank you for your valuable feedback and constructive comments. We greatly appreciate the time and attention you have dedicated to this process.
>
> Sincerely Yours,
>
> The Authors

---

### Meta-Review · Area_Chair_mswN · 2024-12-20

**Metareview:**

The authors propose MiPKD, a multi-granularity mixture of prior knowledge distillation framework designed for image super-resolution tasks. MiPKD facilitates the transfer of “dark knowledge” from teacher models to student models across diverse network architectures. The framework employs feature and block prior mixers to reduce the capacity disparity between teacher and student models for effective knowledge alignment and transfer. Extensive experiments are conducted on three SR models and four datasets, demonstrating that MiPKD significantly surpasses existing KD methods.

**Additional Comments On Reviewer Discussion:**

As all our reviewers agree to accept this paper, my recommendation is "Accept (spotlight)".

---

### Decision · Program_Chairs · 2025-01-22

Accept (Spotlight)